# Quantum barriers engineering toward radiative and stable perovskite photovoltaic devices

Kyung Mun Yeom [1,11], Changsoon Cho [2,3,4,11], Eui Hyuk Jung [5], Geunjin Kim[6], Chan Su Moon[1,6], So Yeon Park [1,7], Su Hyun Kim[1], Mun Young Woo[1], Mohammed Nabaz Taher Khayyat[1], Wanhee Lee[3], Nam Joong Jeon [6], Miguel Anaya [8], Samuel D. Stranks [2,8], Richard H. Friend [2], Neil C. Greenham [2] ✉ & Jun Hong Noh [1,9,10] ✉

Efficient photovoltaic devices must be efficient light emitters to reach the thermodynamic efficiency limit. Here, we present a promising prospect of perovskite photovoltaics as bright emitters by harnessing the significant benefits of photon recycling, which can be practically achieved by suppressing interfacial quenching. We have achieved radiative and stable perovskite photovoltaic devices by the design of a multiple quantum well structure with long (~3 nm) organic spacers with oleylammonium molecules at perovskite top interfaces. Our L-site exchange process (L: barrier molecule cation) enables the formation of stable interfacial structures with moderate conductivity despite the thick barriers. Compared to popular short (~1 nm) Ls, our approach results in enhanced radiation efficiency through the recursive process of photon recycling. This leads to the realization of radiative perovskite photovoltaics with both high photovoltaic efficiency (in-lab 26.0%, certified to 25.2%) and electroluminescence quantum efficiency (19.7 % at peak, 17.8% at 1-sun equivalent condition). Furthermore, the stable crystallinity of oleylammonium-based quantum wells enables our devices to maintain high efficiencies for over 1000 h of operation and >2 years of storage.

Metal-halide perovskite photovoltaic devices (PPVs) are emerging photovoltaic devices, which can challenge or complement Si-based technology. While state-of-the-art single-junction PPVs are already achieving outstanding electrical properties and high power conversion efficiencies (PCEs)[1–7], further improvements toward the detailed balance (DB) limit are achievable only when the photovoltaic devices approach perfect light emitters[8–10]. Through extensive efforts to optimize perovskite crystals and reduce trap densities,

[1]School of Civil, Environmental and Architectural Engineering, Korea University, Seoul, Republic of Korea. [2]Cavendish Laboratory, Department of Physics, University of Cambridge, Cambridge, UK. [3]Department of Materials Science and Engineering, Pohang University of Science and Technology (POSTECH), Pohang, Republic of Korea. [4]Institute for Convergence Research and Education in Advanced Technology, Yonsei University, Seoul, Republic of Korea. [5]Department of Energy Engineering, Korea Institute of Energy Technology (KENTECH), 21 KENTECH-gil Naju, Republic of Korea. [6]Division of Advanced Materials, Korea Research Institute of Chemical Technology (KRICT), Daejeon, Republic of Korea. [7]Chemistry and Nanoscience Center, National Renewable Energy Laboratory, Golden, CO, USA. [8]Department of Chemical Engineering and Biotechnology, University of Cambridge, Cambridge, UK. [9]Department of Integrative Energy Engineering, Korea University, Seoul, Republic of Korea. [10]Graduate School of Energy and Environment (KU-KIST Green School), Korea University, Seoul, Republic of Korea. [11]These authors contributed equally: Kyung Mun Yeom, Changsoon Cho. ✉e-mail: ncg11@cam.ac.uk; junhnoh@korea.ac.kr

electroluminescence (EL) quantum efficiencies (ELQEs) of >10 % are recently being reported in high-efficiency PPVs[2,5,11,12]. Since the previous efforts on PPVs have mostly aimed at efficient light incoupling and charge collection, their device architectures are distinct from those of the state-of-the-art perovskite light-emitting diodes (PeLEDs) designed to maximize light outcoupling.

Here, contrary to the typical consensus, our optical analysis reveals that PPVs with thick perovskites can be even brighter than typical PeLEDs with thin perovskites at the radiative limit, based on the larger benefit of photon recycling. As a practical strategy to achieve such radiative PPVs, we propose a design of perovskite multiple-quantum-wells (MQWs) with thick energy barriers, which can suppress radiation quenching at perovskite interfaces. We could achieve thick interfacial quantum barriers with both high charge selectivity and a stable crystalline phase by adopting long (~3 nm) organic spacers ($L$ cations) of oleylammonium (OLA) molecules. The $L$-site exchange process has enabled precise phase control of the MQWs, which leads to a reasonable charge carrier conductivity, overcoming the previous electrical limitations of those thick barriers. The optical benefits of thick barriers are boosted by the recursive process of photon recycling, yielding radiative PPVs with a high electroluminescence quantum efficiency of 19.7 % at peak and 17.8% at 1-sun equivalent condition. This achieves efficient photovoltaic operation with a high PCE of 26.0% in-lab (certified to 25.2%). Furthermore, our approach effectively mitigates the spontaneous deformation of the existing MQW structure with short $L$ spacers on the 3D perovskite layer by substituting longer $L$-site cations. As a result, the devices with OLA barriers retain 92% of their initial efficiencies after 500 h operation under 1 sun, along with greatly improved air stabilities at room temperature and 60 °C.

## Results and discussion
### Photon recycling changes the design rule for emitters

ELQEs ($\eta_{EL}$) of in PPVs are directly relevant to their non-radiative photovoltage loss ($V_{nr}$), which at open circuit is:

$$V_{nr} = -(k_B T/q) \times \ln\left(\eta_{EL}\left(J_{ph}\right)\right) \quad (1)$$

where the thermal voltage ($k_B T/q$) is 25.9 mV at 300 K, and $J_{ph}$ is the photocurrent density under 1-sun illumination[8,13,14]. To increase ELQEs and approach the DB efficiency limits, not only the internal radiation efficiency ($\eta_{rad}$) of dipoles, but also their external yields of outcoupling must be improved.

The outcoupling efficiency is known to benefit from various optical effects such as photon recycling and microcavity in perovskite optoelectronics. However, their relative contributions have been rarely quantified mainly owing to the difficulties in optical modeling for reabsorbing thin-film emitters such as perovskites. Here, we adopt a recently proposed model[15,16] to resolve the optical divergence arising in reabsorbing emitters. Based on this approach, we could obtain the angular distributions of internal radiation formed in two different perovskite diodes—the one having a thick and rough perovskite (conventional PPVs) and the other having a thin and smooth perovskite (conventional PeLEDs) (Fig. 1a; refer to Methods and Supplementary Figs. 1–2 for the full details). Due to the large refractive index of perovskite ($n_{perov}$ ~ 2.5), only a small fraction of photons within a narrow cone (<23°) can escape the device. The PeLED structure with a thin emissive layer is typically thought to be optimal for light outcoupling, as it confines the emission angle based on the microcavity effect, achieving a direct light extraction efficiency ($F_{out}$) of 17.4 %. This is unlike the PPVs with thick emissive layers where the benefit of optical resonance is diluted over the broad recombination zone and most photons propagate in the lateral modes. That results in a low $F_{out}$ of 2.6 % in PPVs. The rest can have a second chance to be outcoupled if their propagation angle is changed by recursive events of scattering or photon recycling, defined as the reemission of photons reabsorbed by perovskite ($F_{reabs}$), until they are lost through parasitic absorption ($F_{para}$). By considering these effects, the ELQE can be calculated as a function of the $\eta_{rad}$[12,15,17–19]:

$$\eta_{EL} = \eta_{rad} \times (F_{out} + F_{scat})/(1 - \eta_{rad} \times F_{reabs}) \quad (2)$$

where perfect charge balance is assumed and $F_{scat}$ indicates the fraction of photons additionally outcoupled by scattering. Notably, while the $F_{out} + F_{scat}$ of the PPV is still low (4.9 %), the fraction of $F_{reabs}$ is considerably larger in the PPV (88.5 %) than in the PeLED (36.7 %), mainly owing to the thicker perovskite absorber (Supplementary Fig. 3). That results in a significantly reduced $F_{para}$ from 45.9% (in the PeLED) to 6.6% (in the PPV), while photons in the trapped mode get mostly reabsorbed by perovskite before other layers. The reduced optical loss provides more opportunities for photons to be recursively recycled when $\eta_{rad}$ is sufficiently high. Accordingly, while the thin PeLEDs are brighter than the PPV architectures at low $\eta_{rad}$, the ELQE of the PPV rises sharply at high $\eta_{rad}$ (i.e., with more efficient recycling) as shown in Fig. 1b. At the radiative limit ($\eta_{rad}$ = 100%), the ELQE of PPV is

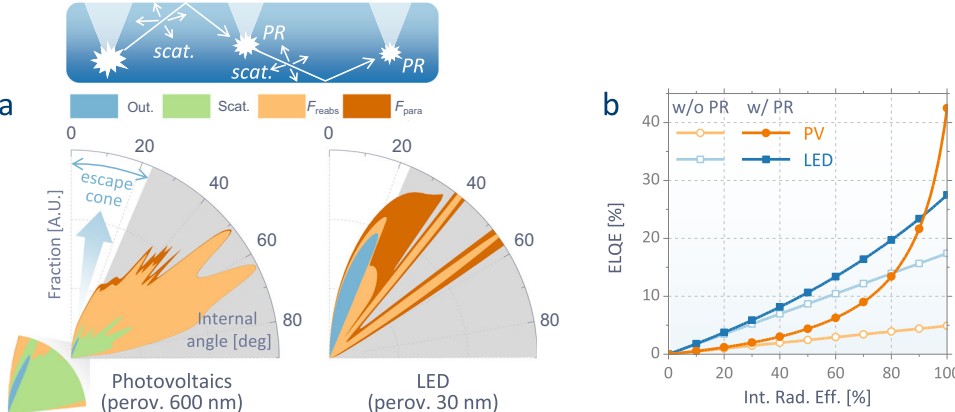

**Fig. 1 | Calculated light emission properties of perovskite devices with varying thickness. a** Angular distributions of internal light emission in conventional PPV (with a 600-nm-thick perovskite and an effective scattering coefficient of $5.6 \times 10^3$ cm⁻¹) and PeLED (with a 30 nm-thick perovskite and no scattering) architectures. Each color represents fractions of optical energy (outcoupled ($F_{out}$), scattered ($F_{scat}$), or reabsorbed by perovskite ($F_{reabs}$) or parasitic layers ($F_{para}$)). The radiated photons can be outcoupled directly through the escape cone or additionally through photon recycling or scattering, as depicted at the top. **b** Calculated ELQE *versus* internal radiation efficiency ($\eta_{rad}$) for both structures with and without the contribution of photon recycling (i.e., $F_{reabs}$ > 0 or $F_{reabs}$ = 0, respectively, in Eq. 2).

predicted to reach 42%, much surpassing that of the PeLED. The result shows that the design rule for efficient LEDs must be changed when perovskites approach the state of an ideal emitter.

Practically, most of the ELQEs currently reported for PPVs are still lower than those of state-of-the-art PeLEDs, due to the insufficient $\eta_{rad}$ in full devices. In photovoltaic operation, an additional challenge arises from the charge-extraction layers that can bring more quenching sites[20,21]. Overcoming such an interfacial quenching is crucial to realize radiative PPVs which can go beyond the ELQE limits of conventional LED architectures and approach the DB efficiency limit.

## Control of interfacial quantum barriers

Organic treatment of three-dimensional (3D) perovskites is a popular strategy used in high-efficiency PPVs. The treatment can produce Ruddlesden–Popper two-dimensional (2D) perovskites with the formula $L_2A_{n-1}B_nX_{3n+1}$ (n: the number of octahedral sheets per layer), forming an MQW structure having octahedral sheets as quantum wells and organic spacers (L cation) as quantum barriers. Previous studies on organic treatment have mostly aimed at passivating trap sites in 3D perovskites or forming a 3D/2D junction to facilitate charge transfer[22–27]. For these purposes, current efficient PPVs are mostly adopting short (~1 nm) L cations such as octylammonium (OA) and phenethylammonium (PEA), which form thin layers and readily conduct charges from 3D perovskites[1–5]. Contrary to such previous designs, we propose an adoption of MQWs with thicker (~3 nm) energy barriers to further suppress interfacial quenching in PPVs and improve the stability. While the poor charge conductance has previously made such thick barriers less popular for high-efficiency devices, we overcome this limitation through the precise phase control of 2D perovskites. Figure 2a illustrates our process of a sequential coating of OAI and oleylamine on the 3D perovskite layer. While OAI treatment forms OA (C8)-based 2D perovskites, oleylamine exchanges its L-site cation with OLA having a longer carbon chain of C18.

For C8 and C18-based 2D perovskites, the X-ray diffraction (XRD) peaks appear at multiples of 3.56° and 2.33°, corresponding to the lattice parameters of 2.5 and 3.8 nm, respectively (Fig. 2b). These peaks are consistent with the OA- and OLA-based Ruddlesden–Popper-phase 2D perovskites with $n = 2$, having ~0.6 nm larger lattice parameters compared to the pure 2D perovskites of $OA_2PbI_4$ and $OLA_2PbI_4$ with $n = 1$, respectively (Supplementary Fig. 4). The result indicates that the C8 spacers of 2D perovskites are effectively substituted by C18 during the process, whereas their octahedral structure remains unchanged. The effectiveness of our process is further supplemented by the XRD analysis of a separately prepared OLA-based 2D perovskite with $n = 2$, exhibiting the same peak position (2.33°) as our C18 MQWs formed on 3D perovskites. (Supplementary Fig. 5) By contrast, direct treatment of OLA iodide on 3D perovskites is known to accompany $n = 1$ crystals[28], which hinders charge conduction much more (Supplementary Figs. 6–7).

With a fixed quantum well thickness of $d_{QW} = 1.2$ nm with $n = 2$, the measured lattice parameters correspond to the barrier thicknesses of $d_{QB} = 1.3$ and 2.6 nm for C8 and C18-based MQWs, respectively. Figure 2c illustrates the calculated probability density ($|\Psi(x)|^2$) of charge carriers in MQWs per energy level, where electrons and holes are transmitted from the left (perovskite) to the right (hole extraction layers) side through the one-dimensional photonic crystals[29]. With C8 barriers, the quantum confinement effect is shown to shift the effective conduction ($E_c$) and valence ($E_v$) bands by +0.74 and −0.44 eV, while C18 barriers additionally shift them by +0.09 and −0.06 eV, respectively. Those shifts widen the effective bandgaps of the interfacial structures, establishing energy barriers and hindering charge carrier transport. This is consistent with the energy levels of these 2D perovskites measured by ultraviolet photoelectron spectroscopy (UPS) and inverse photoelectron spectroscopy (IPES) (Supplementary Fig. 8). The barriers are shown to be higher for electrons than for holes and the

difference between the barriers is larger for C18 ($\Delta E_c - (-\Delta E_v) = 0.33$ eV) than for C8 (0.30 eV). Such a charge selectivity renders the C18-based MQWs suitable for electron blocking layers, preventing non-radiative recombination caused by electrons transmitted from perovskites to the trap states at the hole extraction layers[20,21].

Figure 2d depicts the photoluminescence (PL) of 3D perovskite films with and without MQWs on them. The overall spectra appear to be red-shifted compared to the full device luminescence (Supplementary Fig. 9), whereas both peaks with and without red-shift appear together in Fig. 2e with a hole transporting layer (HTL) contact. The red-shift can be attributed to photons self-filtered during propagation in the waveguide mode[12,15,30]. The red-shift is smaller in full devices in which the guided photons can be lost through parasitic absorption. While our 3D-only perovskites film exhibit a moderately high PL quantum efficiency (PLQE) of 18.5%, C8 and C18 MQWs enhance it to 25.6 % and 21.3 %, respectively. That is consistent with the well-known trap passivation effects of organic treatments[24–27,31–33]. Here, there is no benefit shown for our L-site exchange process in terms of surface passivation, compared to conventional C8-only approaches.

In addition to 3D perovskites, the charge-extraction layers can provide more quenching sites for radiation in full devices[20,21,24]. At the charge-extraction interfaces, in addition to the intrinsic energy bands, defects in the extraction materials or dopants added for charge conduction can induce subgap parasitic energy states, working as non-radiative recombination centers. As indicated in Fig. 2e, the PL of the 3D-only film plummets (PLQE = 0.7%) when the film contacts a doped 2,2′,7,7′-tetrakis[N,N-di(4-methoxyphenyl) amino]−9,9′-spirobifluorene (spiro-OMeTAD), the most popular HTLs in current n-i-p PPVs. The result implies that the charge-extraction layers can be the main source of the non-radiative loss, dominant over the intrinsic trap sites in perovskites. The PL loss can be effectively suppressed by inserting MQWs at the interface (PLQE = 11.2% with C8 and 16.5% with C18), preventing direct contact between 3D perovskites and HTLs. The enhancement is especially larger with thicker barriers of C18 MQWs, differently from the trend shown for neat films in Fig. 2d. Notably, the magnitude of enhancement with MQWs is considerably greater with HTLs than that in neat films, indicating that the optical benefits of MQWs mainly come from the reduced interfacial quenching in charge extraction layers rather than the well-known effects of trap passivation for perovskites. That aspect makes thick C18 MQWs the most beneficial for efficient radiation despite their weaker trap-passivation effects on neat films compared to their C8 counterparts. Supplementary Fig. 10 shows a spatially uniform PL enhancement with C18 over the whole grains. Figure 2f illustrates the PL decay due to charge transfer from perovskite to HTLs. While the valence band of 3D perovskites ($E_v = 5.40$ eV) is better aligned with the highest occupied molecular orbital (HOMO) level of spiro-OMeTAD (5.20 eV, Supplementary Fig. 8), it is shown that MQWs with large bandgap, especially C18 ($E_v = 5.95$ eV), hinder the charge transfer, resulting in slower PL decay. That implies that our strategies with MQWs do not bring an electrical benefit. Figure 2g summarizes the major role of MQWs that we propose, preventing radiation quenching at the interfaces by increasing charge selectivity.

In addition to radiation, C18 MQWs can bring further benefits to the device stability. The XRD patterns in Fig. 2h show that, when conventional C8 MQWs are made on 3D perovskites, they are spontaneously deformed in a week even without external thermal stress. Most 2D perovskites are typically known to be stable in air when they are alone[34], however, their spontaneous deformation on 3D perovskites[35] has been less investigated. Such an unstable crystallinity of C8 MQWs is consistent with the poor device stabilities shown in conventional PPVs with them, as will be discussed later. On the other hand, the XRD peaks of C18 MQWs on 3D perovskites are almost unchanged in the same condition, consistently with other reports[28].

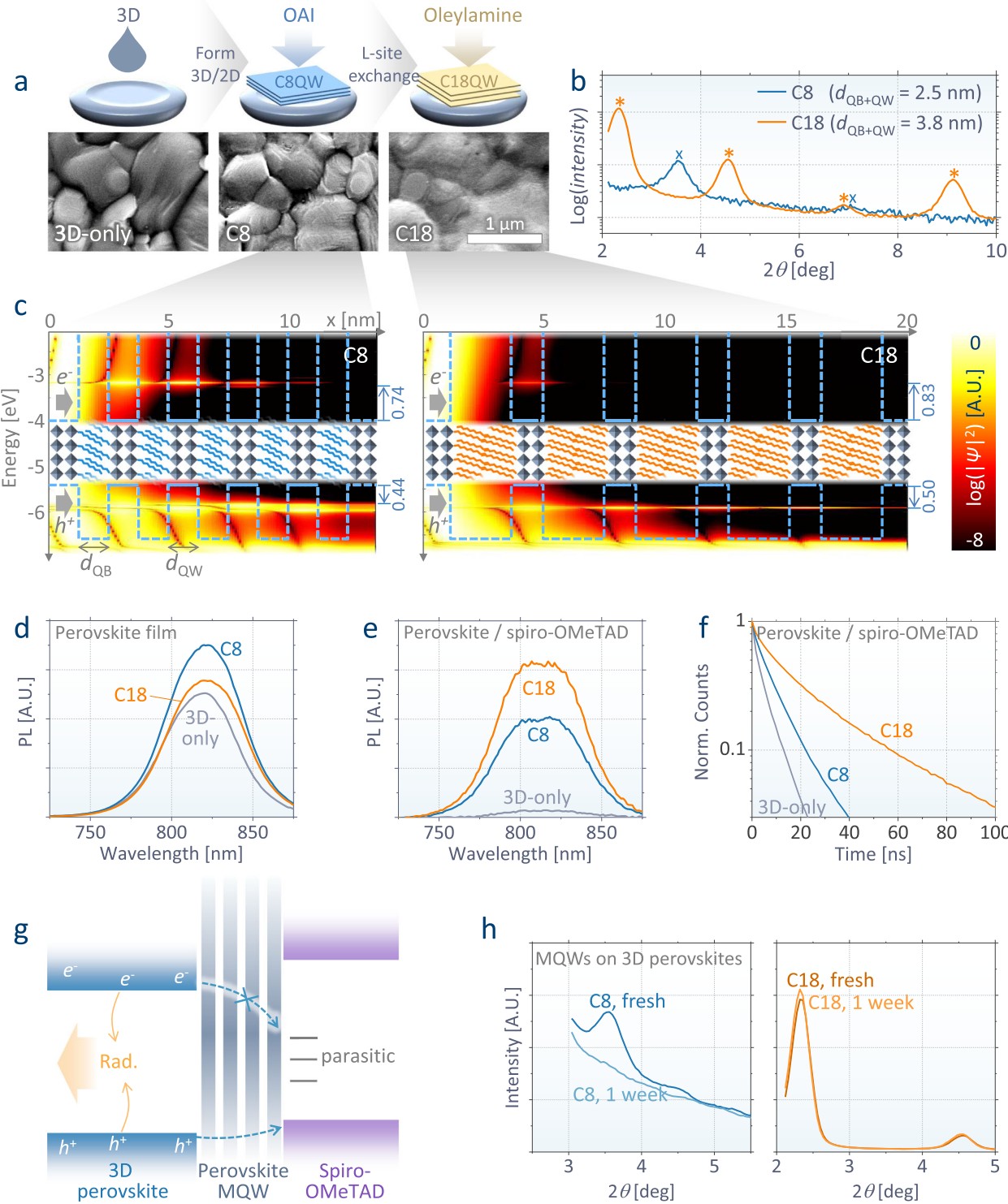

**Fig. 2 | Multiple-quantum-well (MQW) structures of two-dimensional (2D) perovskites based on different organic spacers. a** Scanning electron microscope (SEM) images of three-dimensional (3D) perovskite films before and after sequential coating of octylammonium iodide (OAI) and oleylamine on top. OAI coating on the 3D film forms 2D perovskites ($L_2A_{n-1}B_nX_{3n+1}$) based on OA (C8). OLA (C18) 2D perovskites are formed by exchanging their L-site (OA) with OLA. **b** X-ray diffraction (XRD) patterns of C8 and C18 2D perovskites, indicating the formation of MQW structures with barrier thicknesses ($d_{QB}$) of 1.3 and 2.6 nm, respectively. Both have a well thickness ($d_{QW}$) of 1.2 nm with an octahedral number of $n = 2$. **c** Transfer-matrix calculation for probability density ($|\Psi(x)|^2$, per nm) of electrons ($>-4.0$ eV) and

holes ($<-5.4$ eV) at each energy level injected from $x = 0$ in 5-stacked MQWs with C8 and C18 barriers. The dashed blue lines indicate the input energy bands of the quantum wells and barriers. **d**–**f** Measured spectra (**d**, **e**; under continuous excitation of 0.1 W cm$^{-2}$ at 510 nm ( $\sim$ 0.8 sun)) and transient decays (**f**; under pulsed excitation of 14 nJ cm$^{-2}$ at 470 nm) of the photoluminescence (PL) of those perovskite films without (**d**) and with (**e**, **f**) a spiro-OMeTAD layer on them. **g** A schematic of the perovskite MQWs preventing interfacial non-radiative loss. **h** XRD analysis according to the degradation of C8- and C18-based 2D perovskites on the 3D layer stored in air at room temperature.

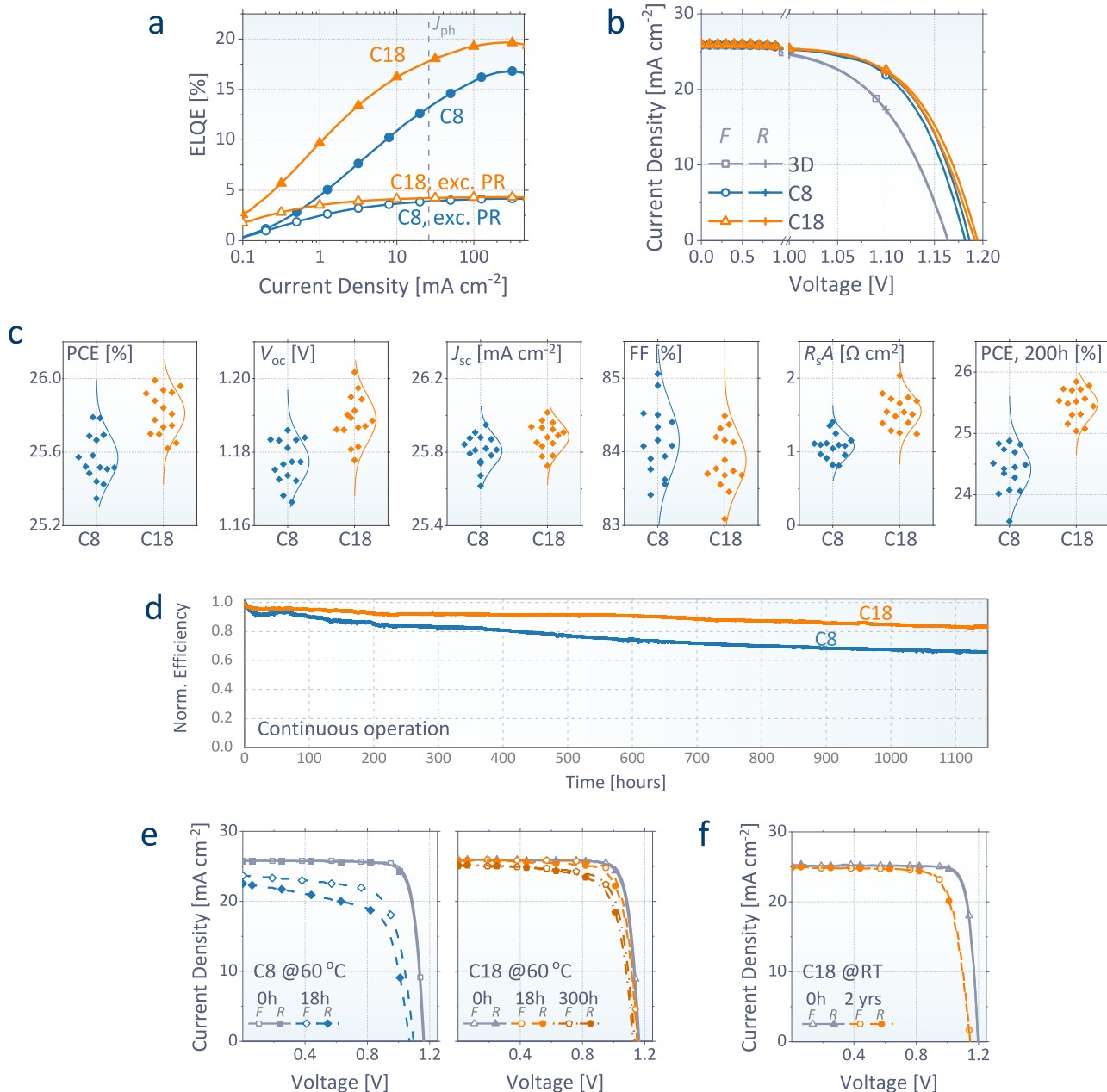

**Fig. 3 | Device performance with conventional (C8) and our (C18) approaches.**
**a** Measured ELQEs of the devices (solid) and corresponding ($F_{out} + F_{scat}$) × $\eta_{rad}$, excluding the contribution of photon recycling (open). **b** Measured current density – voltage (J–V) curves of the champion devices with and without C8 and C18 MQWs (1-sun illumination; forward and reverse scans). **c** Statistics of PCE, $V_{oc}$, $J_{sc}$, FF, $R_sA$, and PCE after 200 h of storage in air without encapsulation. **d** Normalized efficiencies as a function of time for the encapsulated devices under continuous illumination (1 sun) in air tracked at their maximum power point (MPP). Initial PCEs are 25.5% for C18 and 25.1% for C8. For all data, the full devices consist of glass / fluorine-doped tin oxide (FTO) / $SnO_2$ / perovskite / MQW / spiro-OMeTAD / Au, with an antireflection coating on top. **e**, **f** J–V curves after storage at 60 °C (**e**) and that after 2-years storage at room temperature (**f**) both in air without encapsulation.

## Enhanced efficiency and stability of perovskite photovoltaic devices

Figure 3a and Supplementary Fig. 9 illustrate the luminescence performance of PPVs. A device with conventional C8 treatment exhibits a peak ELQE of 16.8 %, consistent with the previous record ELQE (17.2 %) of an OA-based PPV with a similar structure[2]. By adopting our L-site exchange process with C18, that further increases to 19.7%, approaching the record ELQE (22.2%) reported for near-infrared PeLEDs[36]. At an injection of $J_{ph}$ = 26.0 mA cm$^{-2}$, the ELQE increases from 13.5 % with C8 to 17.8 % with C18. Our optical analysis shown in Fig. 1b enabled to distinguish the efficiencies of direct light emission

($\eta_{rad}$ × ($F_{out}$ + $F_{scat}$)) and contribution from photon recycling. In the absence of the benefit of photon recycling, the direct emission efficiencies of our devices are shown to be only 3.9 % ($\eta_{rad}$ of 80%) with C8 and 4.2 % ($\eta_{rad}$ of 86%) with C18. Since our control device already reaches high $\eta_{rad}$, there exists only a small room for further enhancement in the direct emission and the relative enhancement we additionally made in it is only 7.2 %. However, such a small difference results in a larger relative increase (32 %) in their external ELQEs, owing to the nonlinear nature of photon recycling, of which efficiency is proportional to $\eta_{rad}^N$ after the N-th recursion of reabsorption and remission. At the early stage of PPV research, the radiation

efficiencies were low and the contribution of photon recycling was negligible. In that case, the voltage benefit from the 7.2% enhanced emission is marginal (< 2 mV, Eq. 1) and hence, the importance of radiation enhancement has been often underestimated. However, our optical analysis and experimental result quantitatively demonstrate that, even a minor enhancement in internal radiation can substantially benefit the external ELQE and photovoltage. While such an effect is more significant in the devices having a thick perovskite, smaller parasitic absorption, and high $\eta_{rad}$ (Fig. 1), the device deigns for the recent highly efficient and bright PPVs, which mostly satisfy these conditions, must be different from the classical approaches, to maximize the benefits of photon recycling.

The enhanced radiation results in high photovoltaic performance (Fig. 3b; external quantum efficiencies (EQEs) are shown in Supplementary Fig. 11). The OA-based treatment (C8 MQW) has been widely adopted in recent high-efficiency (>25 %) PPVs[2–4] and is effective in our devices as well. Compared to the 3D-only devices, C8 MQWs improve the open-circuit voltage ($V_{oc}$) from 1.164 to 1.184 V and PCE from 24.65 – 25.79 % (champion pixels), which is already among the highest values ever reported for PPVs. In addition, our proposed approach with C18 MQWs further increases the voltage to 1.193 V and leads to the even higher PCE of 26.04 % (short-circuit current density ($J_{sc}$) of 25.98 mA cm$^{-2}$; fill factor ($FF$) of 83.97 %). We shipped a batch of devices to an accredited laboratory (Newport, US) and got a certified PCE of 25.16 % (Supplementary Figs. 12–13). A marginal drop in $FF$s can be attributed to de-doping of dopant ions in spiro-OMeTAD in the quasi-steady-state scan[4,37].

Compared to the previous state-of-the-art technologies with C8 treatment[2–4], the statistics illustrated in Fig. 3c can precisely resolve the benefit of our approach with C18. The insulation with thicker quantum barriers increases the series resistance ($R_sA$) from 1.08 to 1.55 Ω cm2, causing a subtle decrease in $FF$ from 84.15 % – 83.88 %, as predicted from their energy levels (Fig. 2c and Supplementary Fig. 8). In contrast, the average $V_{oc}$ increases from 1.177 to 1.189 V. Supplementary Fig. 14 shows that the voltage benefit of C18 coming from the change in direct emission (($F_{out} + F_{scat}$) × $\eta_{rad}$) is minor, and most of the measured voltage difference originates from the recursive events of photon recycling. Such a boosted optical benefit enables a net increase in the average efficiency from 25.57 % to 25.81 %, overcoming the loss in $FF$. These results validate our proposed design rules with thicker quantum barriers, despite the potential losses in charge transport and trap passivation compared to conventional approaches with thinner barriers.

Consistently with the stable crystallinity shown in Fig. 2h, the benefit of our C18 MQWs is more notable for stabilities. Without encapsulation, the average efficiency of our control devices with C8 rapidly drops to 24.41 % after air storage for 200 h, consistent with the trends shown in other high-efficiency PPVs tested in air[3,4]. On the other hand, the devices with C18 remain at 25.48 % under the same conditions, with a best PCE of 25.85 %. Figure 3d shows the improved photostability with our approach. Our encapsulated C18 devices retained 92% and 83% of their original PCE after 500 and 1,150 h, respectively, under continuous 1 sun illumination. These greatly surpass the 77% and 66% for C8 devices, respectively, in the same condition (Fig. 3d). For thermal stabilities, as indicated in Fig. 3e and Supplementary Fig. 15, the devices with C18 retain 84 % of their initial PCEs after 300 h at 60 °C in air, while those with C8 reduce to 68 % of the initial PCE in only 18 h at 60 °C. Figure 3f and Supplementary Fig. 16 show a long-term air stability of our C18 devices, retaining an $\eta_{EL}(J_{ph})$ of 12% after 2 months and PCE of 22% after 2 years in air.

This approach of thickening the quantum barrier is counterintuitive to traditional device designs due to the low charge conductance of thick barriers. The low conductance has previously limited the application of OLA-based 2D perovskites for highly efficient PPVs, despite their outstanding crystalline stability.

Supplementary Fig. 17 further confirms that coating oleylamine without the L-site exchange process results in a significant electrical loss and low PCE for n-i-p devices. However, for our L-site exchange approach with C18, the electrical loss in $FF$ is relatively small compared to the benefits in $V_{oc}$ and stability for n-i-p devices. Moreover, the L-site exchange process is distinguishable from previous attempts for p-i-n devices based on the direct deposition of OLA iodide on 3D perovskites, which formed mixed phases of $n = 1$ and $2$[28]. Such a reasonable electrical conductance despite the thick barriers can be attributed to the proposed L-site exchange process forming conductive 2D perovskites with $n = 2$.

Based on high ELQEs, our devices with C18 reach 96.4 % of the DB limit of $V_{oc}$, surpassing all other photovoltaic devices and approaching the performance of GaAs[38] (Supplementary Fig. 18). The radiation efficiency reported for PPVs has been rapidly increasing (Supplementary Fig. 19), indicating that we are very close to the point at which PPV architectures will become brighter than conventional LEDs, as predicted in Fig. 1b. Based on the compatibility with recent breakthroughs made in 3D perovskite[3,5] or SnO2 layers[1,2,4], our approaches with MQWs will further accelerate the rise of ELQEs and photovoltaic efficiencies, until their radiative limits are reached. In the future, higher efficiencies of light harvesting and radiation beyond the current radiative limits (ELQE of 42% shown in Fig. 1b) can be also targeted through novel optical designs for device architectures[12,15]. Supplementary Fig. 20 shows that unity ELQE and zero $V_{nr}$ can be approached by reducing the absorption of electrodes in PPVs.

In summary, we adopt thick quantum wells with precisely controlled phases at the perovskite interface through L-site exchange processes with OLA. This approach enhances ELQE and PCE by suppressing charge quenching at the interface in PPVs. This demonstrates the optical benefits outweighing the increase in electrical resistance due to the introduction of thick quantum wells. Furthermore, the stable crystalline structure of the OLA-based MQWs also significantly enhances the photo-, thermal-, and air-stabilities of devices compared to conventional approaches. The design of device architecture that minimizes electrical losses and maximizes optical benefits in PPVs will serve as a promising strategy to approach theoretical efficiency in the future.

## Methods
### Materials
SnCl$_2$ dihydrate, TGA, urea, Zn powder, detergent for an ultrasonic cleaner, MACl, dimethylformamide (DMF), dimethyl sulfoxide (DMSO), oleylamine, octane, chloroform (CF), Bis(trifluoromethane) sulfonimide lithium salt (Li-TFSI), 4-tert-butylpyridine (tBP), acetonitrile (AN), and chlorobenzene (CB) were purchased from Sigma-Aldrich. FAI (FA: formamidinium), MABr (MA: methylammonium), and OAI were purchased from Greatcell Solar. PbI$_2$ and PbBr$_2$ were purchased from TCI. spiro-OMeTAD and tris(2-(1H-pyrazol-1-yl)-4-tert-butylpyridine)cobalt(III)tris(bis(trifluoromethylsulfonyl)imide)) (FK209) were purchased from LumTec. HCl, ethanol, acetone, and isopropyl alcohol (IPA) were purchased from Samchun. Ethyl ether was purchased from Duksan. Fused silica substrates were purchased from Hanjin Quartz. Fluorine-doped tin oxide (FTO) substrates were purchased from Asahi. Au was purchased from iTASCO. All chemicals were used as received without further purification.

### Device preparation
The FTO substrates (2.5 cm × 2.5 cm) were chemically etched using Zn powder and HCl aqueous solution (HCl: distilled water = 1:5 v/v). The etched substrates were ultrasonically cleaned with detergent, IPA, ethanol, and acetone (30 min each), and then treated with a UV–Ozone cleaner for 30 min. SnO$_2$ was deposited with the chemical bath deposition (CBD) method, referring to the steps in a previous report[39]. The substrates were re-treated with the cleaner for another 30 min

before depositing perovskites. A 1.7 M $(FAPbI_3)_{0.97}(MAPbBr_3)_{0.03}$ precursor solution was prepared by dissolving FAI, MABr, $PbI_2$, and $PbBr_2$ in 0.8 ml of DMF and 0.1 ml of DMSO, along with 0.45 M MACl as an additive. The solution was spin-coated at 1000 rpm for 5 s, followed by 5000 rpm for 15 s. At the end of spinning, 1 mL of ethyl ether was poured on the substrate. The films were annealed at 150 °C for 20 min to crystallize the three-dimensional (3D) perovskite phase. Two-dimensional (2D) perovskites were subsequently formed on them by spin-coating OAI (0.0129 g of OAI in 5 ml of CF; 7000 rpm for 30 s; annealing at 100 °C for 13 min) and oleylamine (10 μl of oleylamine in 1 ml of octane; 7000 rpm for 30 s; no annealing) precursor solutions. 1.1 ml of spiro-OMeTAD precursor solution (90.9 mg mL$^{-1}$ in CB) was doped by adding 23 μL of Li-TFSI solution (540 mg mL$^{-1}$ in AN), 10 μL of FK209 solution (376 mg mL$^{-1}$ in AN), and 39 μL of tBP to it. The doped solution was spin-coated on the perovskite films at 2000 rpm for 30 s. Finally, the Au electrode was deposited by thermal evaporation. The cross-section of our full device is shown in Supplementary Fig. 21.

## Device characterization (photovoltaic)

The photovoltaic performance of the devices was measured using a solar simulator (Class AAA-94043A, Newport) under AM 1.5 G illumination, calibrated against a Si-reference cell certificated by the National Renewable Energy Laboratory (NREL), US. The current density vs. voltage ($J − V$) curves were scanned in the forward and reverse directions in the range of -0.20–1.25 V (step of 10 mV; scan rate of 100 mV/s). The illumination area (0.0957 cm$^2$) was confined by a metal mask with an aperture. All measurements were performed at room temperature in air without encapsulation. The devices' air stability was evaluated by placing them without encapsulation in a container containing silica gel at room temperature or 60 °C. Their photo-stability was measured by MPP tracking with encapsulation in air. The EQE was obtained using QUANTX-300 (Newport).

## Device characterization (Electroluminescence)

The current–voltage characteristics of the devices were scanned in the forward direction in a dark room. The response of a pre-calibrated silicon photodiode was simultaneously recorded using two source-meter units and software "SweepMe!". The electroluminescence quantum efficiencies (ELQEs) were calculated from the response of the photodiode assuming a Lambertian angular distribution[40]. The devices were characterized in a setup previously cross-verified against an apparatus of a third-party industrial laboratory[41]. The EL spectrum was obtained using a spectrometer (Ocean Optics).

## Film characterization

The surface morphologies of the film were characterized using a field-emission scanning electron microscopy instrument (Inspect F, FEI). The X-ray diffraction (XRD) spectra were measured using a Rigaku Dmax 2500-PC with an X-ray tube (Cu K$_\alpha$, $\lambda = 1.54$ Å). The energy levels were characterized by UPS and IPES (ULVAC PHI, Japan).

## PL characterization

The photoluminescence (PL) spectra were obtained by exciting the films with a continuous-wave diode laser (510 nm, ~0.1 W cm$^{-2}$). The bare perovskite films (i.e., without spiro-OMeTAD) were encapsulated in a cover glass to avoid direct exposure to the air. The signal was collected using an Andor iDus DU420A Si detector with an integrating sphere. The PLQEs of neat films were quantified according to a previously reported method[42]. Transient PLs were obtained by time-correlated single-photon counting (TCSPC) measurements, equipped with a 470 nm pulsed laser, with a repetition period of 500 kHz, illuminating the back (spiro-OMeTAD) side of the samples. The intensity of the pulsed laser was 14 nJ cm$^{-2}$, corresponding to a charge carrier concentration of $6 \times 10^{14}$ cm$^{-3}$, where bimolecular recombination is sufficiently slower than the charge transfer processes investigated in

this study. We assume that the influence of charge carrier diffusion[43] is marginal to our results of transient PL. Confocal mapping of reflection and PL was performed with a wide-field microscope (IMA VIS$^{TM}$, Photon Etc.)[44]. The reflection maps were obtained with a 100 × objective lens at the film side under white lamp illumination. The PL maps were obtained in the same setup at the same spot by exciting the sample with a wide-field laser (1-sun equivalent; 405 nm; continuous wave) from the substrate side.

## Light emission modeling

The light emission of a full perovskite photovoltaic device (PPV) structure of glass ($n = 1.5$, incoherent) / FTO ($n = 1.9 + 0.007i$, 600 nm) / SnO$_2$ ($n = 1.9$, 80 nm) / perovskite (600 nm) / spiro-OMeTAD ($n = 1.63$, 260 nm) / Au was calculated using a recent method[12,15,16]. The fractions of outcoupling and absorption of each layer were obtained from the Poynting vectors at the interfaces, which were calculated using the transfer-matrix formalism (TMF). The $x$, $y$, and $z$-oriented dipoles were uniformly distributed over the perovskite layer divided into 20 slices. The non-radiative near-field coupling in perovskite was assumed to be fully recycled to avoid divergence in calculations[15,16]. To consider light scattering, a scattering rate ($S_0 = 1/(scattering\ length/2n_{perov}2) = 5.6 \times 10^3$ cm$^{-1}$ for $n_{perov} = 2.56$) was measured using a spectrally resolved PL (Supplementary Fig. 1). Then, the effective scattering coefficient ($k_{scat} = S_0 \times \lambda/4\pi$, $\lambda$: wavelength) was added to the imaginary part of the refractive index of perovskite (Supplementary Fig. 2b), and photons removed through scattering were converted to new random dipoles without changing the wavelength[12]. The re-emission of photons absorbed by perovskite was recursively calculated with an efficiency of $\eta_{rad}$, resetting the spectrum at each event. The charge balance efficiency for electroluminescence (EL) was set to unity. The refractive index of perovskite was measured using ellipsometry (Supplementary Fig. 2b), and those for the others were obtained from the literature[12,45,46]. The internal spectrum of radiation was reversely calculated from the measured external EL spectrum (Supplementary Fig. 2a). The same radiation spectrum and optical constants were used for perovskites in the perovskite light-emitting diode calculation, based on the structure of glass ($n = 1.5$, incoherent) / indium tin oxide (150 nm) / ZnO ($n = 1.9$, 40 nm) / perovskite (30 nm) / organic hole transporting layer ($n = 1.8$, 40 nm)/ MoO$_x$ ($n = 2$, 7 nm) / Au, assuming flat interfaces (i.e., no scattering)[16,47].

## Multiple-quantum-well modeling

The probability density ($|\Psi(x)|2$) of charge carriers in one-dimensional MQW crystals was calculated by a transfer-matrix formalism following a method in the literature[29]. The effective masses of the hole and electron were assumed to be $m^* = 0.15 \times m_0 = 1.37 \times 10^{-31}$ kg[48]. The conduction ($E_c$) and valence ($E_v$) bands of perovskite octahedral sheets (0.6 nm for $n = 1$; 1.2 nm for $n = 2$) were assumed to be -4.00 and -5.40 eV, respectively[49], while those for organic barriers were set to -0.28 and -6.57 eV, respectively[50]. The electrons ($>−4.0$ eV) and holes ($<−5.4$ eV) were injected from $x = 0$ in 5-stacked QWs with C8 or C18 barriers. The calculated $|\Psi(x)|2$ was normalized to make $\int |\Psi(x)|2\ dx = 1$ at each energy level, where the unit is nm$^{-1}$.

## Detailed balance efficiency limits

The detailed balance limits of $V_{oc,DB}$, $J_{sc,DB}$, and $FF_{DB}$ for our PPVs were calculated for a semiconductor bandgap of 1.532 eV and an effective photon flux density of 0.48 sun, considering the ratio of confined illumination area to full device area in practical measurements[8,12]. The $V_{oc}$, $J_{sc}$, and $FF$ of GaAs photovoltaic devices were obtained from the literature[38], and their detailed balance limits were calculated for a bandgap of 1.42 eV. The $V_{oc}/V_{oc,DB}$ and ($J_{sc} \times FF)/(J_{sc,DB} \times FF_{DB})$ for other photovoltaic devices were also obtained from the literature[51,52].

## Reporting summary

Further information on research design is available in the Nature Portfolio Reporting Summary linked to this article.

## Data availability

All data are available in the main text or the supplementary information, and source data underlying this paper are available at https://doi.org/10.6084/m9.figshare.25806874.

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

## Acknowledgements
This work was supported by the National Research Foundation of Korea (NRF) grant funded from the Government of South Korea (NRF-2022M3J1A1085279 and RS-2023-00208467), the Korea Institute of Energy Technology Evaluation and Planning (KETEP) from the Ministry of Trade, Industry & Energy (20214000000680), and the National Research Council of Science & Technology (NST) grant by the Government of South Korea (No. CAP18054-202). This research has been performed as a project NO. KS2422-10 and supported by the Korea Research Institute of Chemical Technology (KRICT). This work was also supported by the EPSRC (EP/S030638/1). S.D.S. acknowledges the Royal Society and Tata Group (UF150033), EPSRC (EP/R023980/1), and the European Research Council under the European Union's Horizon 2020 research and innovation programme (HYPERION, grant agreement no. 756962). M.A. acknowledges funding from the Leverhulme Early Career Fellowship (grant agreement No. ECF-2019-224) funded by the Leverhulme Trust and the Isaac Newton Trust and from the Royal Academy of Engineering under the Research Fellowship programme.

## Author contributions
K.M.Y., C.C., N.C.G., and J.H.N. conceived the project. K.M.Y. fabricated the devices and characterized the photovoltaic properties. K.M.Y. designed perovskite MQWs and analyzed the material properties. C.C. analyzed the optical benefits of MQWs based on optical characterizations and simulations. C.C. also analyzed the photon recycling effect and detailed balance limits, with the input of W.L. E.H.J., G.K., C.S.M., S.H.K. and N.J.J. contributed to optimizing the control devices. G.K., C.S.M., S.Y.P., M.Y.W., and M.N.T.K. contributed to the optimization and analysis of MQWs. M.A. performed PL mapping. C.C. drafted the manuscript with inputs from K.M.Y. S.D.S, R.H.F., N.C.G., and J.H.N. supervised the work and contributed to the revision of the manuscript. All authors participated in the discussion of the results.

## Competing interests
R.H.F. and N.C.G. are founders of a company commercializing perovskite emitters. S.D.S. is a cofounder of Swift Solar. The remaining authors declare no competing interests.
