## [Peer Review File · Nature Communications]

Quantum Barriers Engineering toward Radiative and Stable Perovskite Photovoltaic DevicesREVIEWER COMMENTS

Reviewer #1 (Remarks to the Author):

The paper has now largely improved and I think it could be published after minor revisions. Below, I write down what I would have done differently.

Discussion of Figure 1. The idea is, I think, that the microcavity effect that gives good outcoupling is incompatible with thick emitter layers. Maybe it would be worth adding a sentence of why this is. Is it fundamentally impossible to create a situation, where emission primarily goes into the escape cone but the absorber layer is still thick?

Figure 1, limit of $\eta_{\text{rad}}=1$. In this limit, ELQE approaches the ratio of outcoupling/(outcoupling+parasitic absorption) as there is nothing else that can happen. The thick layer leads to less outcoupling, if parasitic absorption was assumed constant, the thick layer would always be worse. Why is it better? It must mean, parasitic absorption becomes much less for the thick film. How can we understand this? Let us make an example. Would it be possible to have a figure with the three fractions parasitic absorption, reabsorption within the active layer and outcoupling (sum of LEE_0 and S) as a function of thickness? I think that the thicker you make it, the more you suppress parasitic absorption in % of total emission events and that is key at least for my understanding. In the absence of parasitic absorption, there would be no difference between thick and thin at least in the high η_{rad} limit. That also means that the better the internal yield becomes, the more it will be important to identify the sources of parasitic absorption. This is a well-known concept from GaAs single junction PV, where the introduction of the thin-film GaAs devices by Alta led to highly improved Voc values. So if you were able to say more about the main sources of parasitic absorption in the simulations that would be helpful.

Regarding the relation between parasitic absorption and Voc or ELQE, I think the reference list is somewhat unbalanced. There are many self-citations (e.g. 12, 15, 17) but little in terms of other people's work on the topic, which does exist. There is e.g.

<https://journals.aps.org/prb/abstract/10.1103/PhysRevB.98.075141>

<https://pubs.acs.org/doi/abs/10.1021/acsenergylett.6b00223>

Minor things: ELQE is not an appropriate symbol appearing in an equation according to mathematic conventions of notation. If it is roman, it would have to be a unit (which it isnt) so it has to be italic. If it is italic, however, it means $E^*L^*Q^*E$. So, I suggest to use something else. Every parameter of a certain type should have a single greek or latin letter with indices further specifying its meaning. The problem that I

see is that many people are doing physics that never publish in physics journals anymore (that would enforce this).

The rest of the paper is also not properly formatted in terms of variables and equations. Examples are LEE_0 and S. They should both use the same letter as A_act or A_para to clarify that they are the same type of parameter (a probability). If you choose to use (italic) A for probabilities (which is extremely non-intuitive, but your choice), then LEE_0 and S should also be A_someindex.

Reviewer #3 (Remarks to the Author):

The authors have addressed all my concerns and I think it can be polished in Nature Communications.

Reviewer #4 (Remarks to the Author):

I have read the original manuscript, the authors' response to referees, and the revised manuscript. I apologize for the long time that it has taken because there are lots of materials to read and disgust. I fully agree that the device engineering is very impressive in this work but it seems that there is a lack of novel physics for publication in nature. But I feel this work is certainly suitable for nature communications. A few suggestions for the authors:

1. I'm not sure if the lengthy discussion about the difference between PPV and PLeD is necessary because the key message of this work is very simple -- the authors find a passivation method (using C18) to reduce the non-radiative recombinations so even the film is thick the light emission is still efficient, because of photon recycling (BTW, PR is already well-known in III-V and PVSK PV devices and there is no much new on the physics front). A slightly confusing part is that the authors assume a flat film with a lower thickness for LED, but in reality the most efficient LEDs are based on island structures (e.g., Nature 562, 249–253 (2018); Nature 615, 830–835 (2023)). The outcoupling and ELQE are eventually very much dependent on the geometry and morphology of the devices. Therefore, I am not sure if the discussion related to Fig 1 is necessary.

2. Regarding the use of C18 instead of C8, I certainly understand C18 is longer and can improve stability. But it is not very clear why it can better passivate the 3D perovskite. Also, what is the thickness of this C18 layer? Is there any optical effect to trap the light more (or less) efficiently?

3. To more rigorously confirm the formation of $n=2$ phase on the 3D perovskite, single crystal XRD of the $n=2$ should be obtained and compared with the powder XRD obtained from the 2D/3D film. This is less critical given the nature of this paper is more on the device engineering. But it could be much better if the real crystal structure can be resolved and confirmed before a novel structure is being claimed.

Response to Reviewers' Comments and Summary of Changes

General:

We thank the reviewers for their comments and advice. We thank reviewer #3 for his/her acceptance of our revised manuscript, and also thank reviewers #1 and #4 for their positive opinions and additional suggestions. We have revised the manuscript as follows to reflect the reviewers' comments. We hope that the revised manuscript will satisfy all reviewers.

Reviewer #1's comments and authors' responses:

The paper has now largely improved and I think it could be published after minor revisions. Below, I write down what I would have done differently.

We thank the reviewer for taking valuable time to participate in the review. We hope further revisions will satisfy the reviewer.

Discussion of Figure 1. The idea is, I think, that the microcavity effect that gives good outcoupling is incompatible with thick emitter layers. Maybe it would be worth adding a sentence of why this is. Is it fundamentally impossible to create a situation, where emission primarily goes into the escape cone but the absorber layer is still thick?

R 1.1) The idea is based on the design rules for thin-film LEDs that position the dipoles at the specific resonant spots where the microcavity effect is maximised. In thick films, the recombination zone is broad and such a resonant effect is easily diluted. But fundamentally, the situation that the reviewer mentioned can happen if we can narrow down the recombination zone so the dipole positions get more controllable. We update the manuscript based on the reviewer's suggestion, as shown below:

“The PeLED structure with a thin emissive layer is typically thought to be optimal for light outcoupling, as it confines the emission angle based on the microcavity effect, achieving a direct light extraction efficiency (F_{out}) of 17.4 %. This is unlike the PPVs with thick emissive layers where the benefit of optical resonance is diluted over the broad recombination zone and most photons propagate in the lateral modes. That results in a low F_{out} of 2.6 % in PPVs.” (page 4)

Figure 1, limit of $\eta_{rad}=1$. In this limit, ELQE approaches the ratio of outcoupling/(outcoupling+parasitic absorption) as there is nothing else that can happen. The thick layer leads to less outcoupling, if parasitic absorption was assumed constant, the thick layer would always be worse. Why is it better? It must mean, parasitic absorption becomes much less for the thick film. How can we understand this? Let us make an example. Would it be possible to have a figure with the three fractions parasitic absorption, reabsorption within the active layer and outcoupling (sum of LEE_0 and S) as a function of thickness? I think that the thicker you make it, the more you suppress parasitic absorption in % of total emission events and that is key at least for my understanding. In the absence of parasitic absorption, there would be no difference between thick and thin at least in the high η_{rad} limit. That also means that the better the internal yield becomes, the more it will be important to identify the sources of parasitic absorption. This is a well-known concept from GaAs

single junction PV, where the introduction of the thin-film GaAs devices by Alta led to highly improved Voc values. So if you were able to say more about the main sources of parasitic absorption in the simulations that would be helpful.

R 1.2) We agree the points that the reviewer raised and thank for the suggestions.

First, as the reviewer pointed out, the maximum EQE is simply determined by the ratio between direct outcoupling and parasitic absorption. Hence, the higher EQE for solar cells can be attributed to the significant reduction in parasitic absorption, which was not sufficiently clarified in the previous version of our manuscript. The reduction in parasitic absorption comes from the increased perovskite reabsorption in the trapped mode, as shown in Figure 1a. Since most of the photons in the waveguide mode in the PPV are reabsorbed by a thick perovskite before getting absorbed by other layers, the relative fraction of parasitic absorption becomes very small in the PPV. While perovskite reabsorption can reduce both outcoupling and parasitic absorption, since the internal optical path length is much longer for the waveguide mode (typically several micrometres or longer) than for the air mode (shorter than perovskite thickness), the relative reduction is more significant for parasitic absorption than for outcoupling. We have updated the manuscript by introducing it, as shown below.

“Notably, while the $F_{\text{out}} + F_{\text{scat}}$ of the PPV is still low (4.9 %), the fraction of F_{reabs} is considerably larger in the PPV (88.5 %) than in the PeLED (36.7 %), **mainly** owing to the thicker perovskite absorber (Supplementary Figure 3). **That results in a significantly reduced F_{para} from 45.9% (in the PeLED) to 6.6% (in the PPV), while photons in the trapped mode get mostly reabsorbed by perovskite before other layers. The reduced optical loss** provides more opportunities for photons to be recursively recycled when η_{rad} is sufficiently high. Accordingly, while the thin PeLEDs are brighter than the PPV architectures at low η_{rad} , the ELQE of the PPV rises sharply at high η_{rad} (i.e., with more efficient recycling) as shown in Figure 1b.” (page 5)

Moreover, the different charge transporting layers and electrodes of the PPV and PeLED structures that we assumed may complicate the analysis for their different optical properties. To better distinguish the impact of perovskite thickness, we added the simulation data for varying perovskite thickness in PPVs maintaining all other layers, as the reviewer suggested. Please refer to our new Supplementary Figure 3 shown below.

Regarding the relation between parasitic absorption and V_{oc} or $ELQE$, I think the reference list is somewhat unbalanced. There are many self-citations (e.g. 12, 15, 17) but little in terms of other people's work on the topic, which does exist. There is e.g.

<https://journals.aps.org/prb/abstract/10.1103/PhysRevB.98.075141>

<https://pubs.acs.org/doi/abs/10.1021/acsenergylett.6b00223>

R 1.3) We added the references that the reviewer suggested. Please refer to the updated manuscript.

*Minor things: $ELQE$ is not an appropriate symbol appearing in an equation according to mathematic conventions of notation. If it is roman, it would have to be a unit (which it isnt) so it has to be italic. If it is italic, however, it means $E*L*Q*E$. So, I suggest to use something else. Every parameter of a certain type should have a single greek or latin letter with indices further specifying its meaning. The problem that I see is that many people are doing physics that never publish in physics journals anymore (that would enforce this).*

The rest of the paper is also not properly formatted in terms of variables and equations. Examples are LEE_0 and S . They should both use the same letter as A_{act} or A_{para} to clarify that they are the same type of parameter (a probability). If you choose to use (italic) A for probabilities (which is extremely non-intuitive, but your choice), then LEE_0 and S should also be $A_{someindex}$.

R 1.4) We thank the reviewer for the suggestions. While A of A_{act} and A_{para} indicated absorption in the

original manuscript, here, we chose F (fraction) to integrate all the variables' names beyond the absorptions. We have updated the symbols as following:

$$ELQE \rightarrow \eta_{EL}$$

$$LEE_0 \rightarrow F_{out}$$

$$S \rightarrow F_{scat}$$

$$A_{act} \rightarrow F_{reabs}$$

$$A_{para} \rightarrow F_{para}$$

Please refer to our updated manuscript.

Reviewer #3's comments and authors' responses:

The authors have addressed all my concerns and I think it can be published in Nature Communications.

We thank the reviewer for helpful suggestions in the previous round and his/her recommendation of publication.

Reviewer #4's comments and authors' responses:

I have read the original manuscript, the authors' response to referees, and the revised manuscript. I apologize for the long time that it has taken because there are lots of materials to read and disgust. I fully agree that the device engineering is very impressive in this work but it seems that there is a lack of novel physics for publication in nature. But I feel this work is certainly suitable for nature communications. A few suggestions for the authors:

We thank the reviewer for taking valuable time to participate in the review. We have revised the manuscript to reflect your suggestions. We hope these modifications satisfy the reviewer.

1. I'm not sure if the lengthy discussion about the difference between PPV and PLeD is necessary because the key message of this work is very simple -- the authors find a passivation method (using C18) to reduce the non-radiative recombinations so even the film is thick the light emission is still efficient, because of photon recycling (BTW, PR is already well-known in III-V and PVSK PV devices and there is no much new on the physics front).

R 4.1.1)

We agree that some readers of this paper may be more focused on device engineering itself rather than our detailed discussion of photon recycling. While we respect the reviewer's viewpoint, on the other hand, we would like to get the reviewer's understanding of the situation that our manuscript has been already reviewed by other three reviewers and Figure 1 was among the critical parts of the review. Especially Reviewer 1 picked this result as his/her favourite part in our manuscript (refer to the first review in *Nature*) and we added more discussion about it following his/her comments in this round. While the theory of photon recycling itself is already well known as the reviewer pointed out, their practical benefit in real devices has been rarely quantified, mainly owing to the difficulties in optical modelling for reabsorbing emitters such as perovskites. In conferences, we have also seen considerable demand on such a study, from the people who are interested in photonic designs for future solar cells and LEDs beyond the boundary of materials science.

Moreover, in our manuscript, photon recycling is the key to explanation for largely improved V_{oc} with C18, despite the apparently small room for improvement existing in the C8-based solar cells. Our achievement of high EQE_{EL} (19.7%) would be evaluated differently depending on whether its upper bound is 100% (purely theoretical), 42% (that we quantified), or ~20% (of typical good NIR PeLEDs). These are not distinguishable without quantification of various internal optical modes (perovskite reabsorption, parasitic absorption, etc.) discussed in Figure 1.

We have updated our manuscript to clarify how the discussion in Figure 1 is related to our main story, as shown below, in addition to the updates made in R1.1 and R1.2. We hope our updated manuscript can satisfy all the reviewers having diverse viewpoints on this part.

“To increase ELQEs and approach the DB efficiency limits, not only the internal radiation efficiency (η_{rad}) of dipoles, but also their external yields of outcoupling must be improved. The outcoupling efficiency is known to benefit from various optical effects such as photon recycling and microcavity in perovskite optoelectronics. However, their relative contributions have been rarely quantified mainly owing to the difficulties in optical modelling for reabsorbing thin-film emitters such as perovskites. Here, we adopt a recently proposed model^{15,16} to resolve the optical divergence arising in reabsorbing emitters. Based on this approach, we could obtain the angular distributions of internal radiation formed in two different perovskite diodes – the one having a thick and rough perovskite (conventional PPVs) and the other having a thin and smooth perovskite (conventional PeLEDs) (Figure 1a; refer to Methods and Supplementary Figures 1-2 for the full details).” (page 4)

“Our optical analysis shown in Figure 1b enabled to distinguish the efficiencies of direct light emission ($\eta_{\text{rad}} \times (F_{\text{out}} + F_{\text{scat}})$) and contribution from photon recycling. In the absence of the benefit of photon recycling, the direct emission efficiencies of our devices are shown to be only 3.9 % (η_{rad} of 80%) with C8 and 4.2 % (η_{rad} of 86%) with C18. Since our control device already reaches high η_{rad} , there exists only a small room for further enhancement in the direct emission and the relative enhancement we additionally made in it is only 7.2 %. However, such a small difference results in a larger relative increase (32 %) in their external ELQEs, owing to the nonlinear nature of photon recycling, of which efficiency is proportional to η_{rad}^N after the N -th recursion of reabsorption and remission.” (page 12)

“However, our optical analysis and experimental result quantitatively demonstrate that, even a minor enhancement in internal radiation can substantially benefit the external ELQE and photovoltage. While such an effect is more significant in the devices having a thick perovskite, smaller parasitic absorption, and high η_{rad} (Figure 1), the device designs for the recent highly efficient and bright PPVs, which mostly satisfy these conditions, must be different from the classical approaches, to maximise the benefits of photon recycling.” (page 12)

A slightly confusing part is that the authors assume a flat film with a lower thickness for LED, but in reality the most efficient LEDs are based on island structures (e.g., *Nature* 562, 249–253 (2018); *Nature* 615, 830–835 (2023)). The outcoupling and ELQE are eventually very much dependent on the geometry and morphology of the devices. Therefore, I am not sure if the discussion related to Fig 1 is necessary.

R 4.1.2)

As the reviewer pointed out, island nanostructures are currently popular for NIR LEDs, while planar structures are still popular for most visible LEDs. Here we chose a planar structure to generalise our analysis and focus on the impact of perovskite thickness itself, excluding other side effects arising from specific structures. To avoid a confusion, we have revised the expressions as shown below.

“Based on this approach, we could obtain the angular distributions of internal radiation formed in two different perovskite diodes – the one having a thick and rough perovskite (conventional PPVs) and the other having a thin and smooth perovskite (conventional PeLEDs) (Figure 1a; refer to Methods and Supplementary Figures 1-2 for the full details).” (page 4)

While a few of us have contributed to some of the works referred by the reviewer (e.g., *Nature* 615, 830 (2023)), we also have a follow-up research data to be soon published elsewhere, quantifying the optical outcoupling effects of these island structures (Cho, Sun, and Greenham *et al.*, *submitted*). Contrary to conventional belief, the upper limit EQEs for the island structured devices are calculated to be similar to (or even slightly lower than) those for the planar devices presented here (~27%). That is because the planar devices largely benefit from the microcavity resonance, which is reduced in non-planar structures. The scattering effects arising from typical islands with 30-50 nm height are calculated to be not strong enough to go beyond the microcavity effect, as shown in the figure attached below (unpublished). Hence, this issue does not change the main conclusion of our discussion in Figure 1. While the detailed LED optics for island structures may deviate the scope of this research, we believe our Figure 1 will provide helpful insights to the researchers who are interested in optical designs for solar cells and LEDs, regardless of the detailed morphologies.

<Our simulation data submitted to elsewhere>

2. Regarding the use of C18 instead of C8, I certainly understand C18 is longer and can improve stability. But it is not very clear why it can better passivate the 3D perovskite.

R 4.2.1)

As the reviewer pointed out, the long organic chains do not have a benefit in conventional trap passivation, compared to short chains. Also in our Figure 2d, the passivation effect of our C18 is even worse than conventional C8. Instead of trap passivation in 3D perovskites, C18-based 2D perovskites improve the charge selectivity at the interface based on thicker quantum barriers shown in Figure 2c. Hence the interface quenching is largely reduced when spiro-OMeTAD is coated on perovskites (Figure 2e-g). This is our design rule for interfacial layers, distinguishable from typical approaches focusing on trap passivation effects. We updated the manuscript to better clarify our main point as shown below.

“That is consistent with the well-known trap passivation effects of organic treatments.”^{22-25,29-31}

Here, there is no benefit shown for our L-site exchange process in terms of surface passivation, compared to conventional C8-only approaches.” (page 10)

“Figure 2g summarizes the major role of MQWs that we propose, preventing radiation quenching at the interfaces by increasing charge selectivity.” (page 11)

“Such a boosted optical benefit enables a net increase in the average efficiency from 25.57 % to 25.81 %, overcoming the loss in *FF*. These results validate our proposed design rules with thicker quantum barriers, despite the potential losses in charge transport and trap passivation compared to conventional approaches with thinner barriers.” (page 14)

Also, what is the thickness of this C18 layer? Is there any optical effect to trap the light more (or less) efficiently?

R 4.2.2)

We appreciate the reviewer's valuable questions. Reflecting the reviewer's opinions, we have re-investigated the thickness of the C18 layer. The layer was shown to be very thin and not well resolvable in the scanning electron microscope (SEM) analysis, as shown in Supplementary Figure 21. The clearer image could be obtained by the scanning transmission electron microscopy (STEM) analysis after a focused ion beam (FIB) process. The STEM has provided a higher resolution image for the C18-based MQW (attached below), confirming its very small thickness, which is almost ignorable compared to the morphology of perovskite or spiro-OMeTAD. However, this image only illustrates that the C18-based MQW (N2) layer is extremely thin, measuring only tens of nanometers. But, obtaining a very clear image including the boundaries of the C18-based MQW (N2) layer proved to be difficult. Precise analysis, minimizing damage from ion and electron beams during FIB processing and TEM analysis, is required to achieve clearer images. This paper firstly reports on the fabrication method of C18-based MQW (N2) layers and focuses on their advantages from a device perspective compared to the existing C8-based MQW layers. Therefore, as a follow-up study to this paper, we look forward to more precise subsequent research such as electron diffraction in high-resolution transmission electron microscopes (HRTEM) conducted without causing damage. Through such investigations, we anticipate elucidating the crystallographic details and optical effects of these layers.

<Bright field scanning transmission electron microscopy (STEM) cross-section image of 3D Perovskite/C18 layer/Spiro-OMeTAD structure. While the image resolution is limited by a focused ion beam process damaging on thin-films, the thin C18 layer is distinguishable at the interface between the layers of spiro-OMeTAD and 3D perovskite.>

3. To more rigorously confirm the formation of $n=2$ phase on the 3D perovskite, single crystal XRD of the $n=2$ should be obtained and compared with the powder XRD obtained from the 2D/3D film. This is less critical given the nature of this paper is more on the device engineering. But it could be much better if the real crystal structure can be resolved and confirmed before a novel structure is being claimed.

R 4.3)

We respect the reviewer's valuable suggestions. Based on the reviewer's suggestion, we have additionally prepared an OLA-based 2D perovskite crystals with $n = 2$ and added its XRD data to the manuscript, as shown below. Although we were not able to obtain the single crystal data in a publishable quality due to limited synthesis techniques, instead, we could supplement our main claim by measuring a 2D polycrystalline perovskite based on the precisely controlled precursor composition. The XRD peak of our $\text{OLA}_2\text{FAPb}_2\text{I}_7$ ($n = 2$) sample (2.33°) is shown to be well consistent with the peak observed in our C18 MQWs formed on the 3D film (2.33°) shown in Figure 2b. This experimentally demonstrates that MQWs based on $\text{OLA}_2\text{FAPb}_2\text{I}_7$ crystals with an octahedral sheet number of 2 are formed through the designed L-site exchange process.

We have also updated the manuscript accordingly, as shown below.

“The result indicates that the C8 spacers of 2D perovskites are effectively substituted by C18 during the process, whereas their octahedral structure remains unchanged. **The effectiveness of our process is further supplemented by the XRD analysis of a separately prepared OLA-based 2D perovskite with $n = 2$, exhibiting the same peak position (2.33°) as our C18 MQWs formed on 3D perovskites.**” (page 9)

Lastly, we will appreciate the reviewer’s understanding that this paper focuses on forming 2D perovskite MQWs with controlled octahedral number via our unique L-site exchange method on 3D films and reporting its impact in the solar cell devices. We hope our extra efforts make the reviewer satisfied.

REVIEWERS' COMMENTS

Reviewer #1 (Remarks to the Author):

The reviewers have replied to all my comments and I have no further concerns.

Reviewer #4 (Remarks to the Author):

The authors have addressed my questions and made further revisions. In my opinion this work can be accepted now.